# Innate Sensing of Viral Nucleic Acids and Their Use in Antiviral Vaccine Development

**DOI:** 10.3390/vaccines13020193

**Published:** 2025-02-16

**Authors:** Takuji Enya, Susan R. Ross

**Affiliations:** Department of Microbiology and Immunology, University of Illinois at Chicago College of Medicine, Chicago, IL 60612, USA; enya0129@uic.edu

**Keywords:** innate immune system, cytosolic sensors, vaccine, pattern recognition receptors, adjuvant, mRNA cap structure

## Abstract

Viruses pose a significant threat to humans by causing numerous infectious and potentially fatal diseases. Understanding how the host’s innate immune system recognizes viruses is essential to understanding pathogenesis and ways to control viral infection. Innate immunity also plays a critical role in shaping adaptive immune responses induced by vaccines. Recently developed adjuvants often include nucleic acids that stimulate pattern recognition receptors which are essential components of innate immunity necessary for activating antigen-presentation cells and thereby bridging innate and adaptive immunity. Therefore, understanding viral nucleic acid sensing by cytosolic sensors is essential, as it provides the potential means for developing new vaccine strategies, including effective adjuvants.

## 1. Introduction

During virus infection, viral nucleic acids that are both released from virions during capsid disintegration and generated upon replication in host cells are recognized by specific receptors termed pattern recognition receptors (PRRs). PRRS play a critical role in detecting viral nucleic acids and, when activated, induce the production of type I interferons (IFNs) and cytokines, resulting in the expression of IFN-stimulated genes (ISGs) in sentinel cells such as monocytes and dendritic cells [1]. This early immune activation subsequently facilitates adaptive immune responses, including antigen trafficking to the lymphoid organs, antigen presentation, and the activation of T and B lymphocytes. The process is fundamental to the generation of virus-specific immunological memory, enhancing long-term protection against future infections.

Live attenuated and inactivated virus vaccines contain many of the same nucleic acid and protein molecules as replication-competent viruses and thus also have the potential to trigger innate immune responses and contribute to adaptive immunity. Moreover, the recently developed mRNA vaccines leverage PRR recognition mechanisms to induce effective immune responses [2]. These vaccines introduce RNA encoding viral proteins into host cells, where PRR recognition leads to IFN-I and cytokine production along with the biosynthesis of viral antigens [3,4,5,6]. This unique feature of mRNA vaccines—their ability to simultaneously activate both innate and adaptive immune responses—is a key factor in their rapid and effective induction of protective immunity.

Currently, attention is directed towards adjuvants that enhance PRR activation, facilitating robust immune responses upon vaccination [7,8]. This strategy offers the potential for more rapid and potent immune protection against viral pathogens. Thus, understanding the cellular recognition of viral nucleic acids and the subsequent signal transduction is essential for improving the design and efficacy of antiviral vaccines. Several excellent reviews that cover PRR recognition of pathogen molecules other than nucleic acids have recently been published [7,8]. In this review, we present a brief overview of the factors involved in recognizing viral nucleic acids and their signaling pathways and describe recent advances in the use of such antiviral responses in vaccine development.

## 2. The Role of Nucleic Acid Sensing in Pathogen Recognition and Immune Response

The innate immune system serves as a first line of defense against pathogenic infections by employing an array of PRRs and associated signaling pathways that detect the pathogen-associated molecular patterns (PAMPs) found in the various nucleic acids generated during virus infection, including single-stranded RNA (ssRNA), single-stranded DNA (ssDNA), RNA–DNA hybrids, double-stranded RNA (dsRNA), and double-stranded DNA (dsDNA). These PRRs, which include Toll-like receptors (TLRs), NOD-like receptors (NLRs), the retinoic acid-inducible gene I-like receptors (RLRs) retinoic acid-inducible gene-I (RIG-I) and melanoma differentiation-associated protein-5 (MDA5), absent in melanoma 2 (AIM2)-like receptors (ALRs), cyclic GMP–AMP synthase (cGAS), and receptors with DEAD box helicase (DDX) and zinc finger (ZNF) domains, are evolutionarily conserved and play an essential role in mounting a rapid innate immune response [1,9,10,11,12,13] (Figure 1).

Once PAMPs are recognized, PRR signaling initiates the production of IFNs, pro-inflammatory cytokines, and chemokines, which in turn activate transcription of additional antiviral genes. For example, type I IFNs (IFN-I) bind to the IFN α/β receptor (IFNAR), which activates Janus kinases (JAKs). JAKs then phosphorylate and activate the transcription factors signal transducer and activator of transcription 1 (STAT1) and STAT2, leading to the expression of IFN-stimulated genes (ISGs) [14,15,16] (Figure 1).

Pathogens infect different cell types and tissues, and many replicate in distinct cellular compartments. This, and the need for targeted responses to different pathogens, has likely led to the expansion of PRRs with different signaling pathways. These specific sensing and signaling pathways are also important targets for virus proteins that counteract their action and cause host immune evasion [1,17]. As a result, this greatly accelerates host–pathogen co-evolution [18,19,20,21].

## 3. TLRs in Innate Immunity: Detecting Pathogens and Triggering Immune Signaling Pathways

TLRs are a conserved family of type I transmembrane proteins that play crucial roles in innate immunity by recognizing a variety of PAMPs, thereby initiating immune responses against a broad spectrum of pathogens [9,13]. TLRs are strategically located either on the cell surface or within intracellular compartments such as endosomes and lysosomes. For example, TLR1, 2, 4, 5, 6, and 11 are primarily responsible for recognizing extracellular microbial components, while TLR3, 7, 8, and 9 recognize viral or bacterial nucleic acids within endosomal compartments. Upon ligand binding, TLRs undergo dimerization, forming either homodimers or heterodimers, which then trigger intracellular signaling cascades through their Toll/IL-1 receptor (TIR) domain. This interaction recruits specific adaptor proteins, activating downstream pathways such as the NF-κB signaling axis or the TBK1 pathway. These cascades culminate in the activation of transcription factors, including NF-κB, IRF3 and 7, driving the production of pro-inflammatory cytokines and IFN-I essential for mounting effective immune responses.

TLRs act as PRRs for a wide variety of PAMPs. TLR2 forms heterodimers with either TLR1 or TLR6, recognizing structurally diverse bacterial lipoproteins. TLR4, while primarily recognizing LPS from Gram-negative bacteria, also detects components from certain parasites and fungi, as well as proteins derived from viruses and host cells (see below) [9,13,22]. TLR5 binds to flagellin, a protein found in bacterial flagella, and TLR11 is known to recognize profilin-like proteins derived from protozoa [9,13,23]. TLR2 also interacts with viral proteins as a TLR2/6 heterodimer, mediating cytokine production during infection with some strains of lymphocytic choriomeningitis virus (LCMV), New World arenaviruses, measles virus, respiratory syncytial virus, and herpes simplex virus (HSV)-1 [24,25,26,27,28].

TLRs 3, 7, and 9 are important sensors of viral nucleic acids. TLR3 recognizes dsRNA, triggering immune responses in response to West Nile virus (WNV) and influenza A virus (IAV), as well as HSV [29,30,31,32,33]. TLR7 detects viral ssRNA, inducing the production of type I interferons and inflammatory cytokines during infections with IAV, vesicular stomatitis virus, and WNV, as well as in response to human immunodeficiency virus (HIV)-1 ssRNA transfection [34,35,36,37]. TLR9, the first identified DNA sensor, recognizes unmethylated CpG DNA as its ligand, participating in immune responses to bacterial and viral DNA, and plays a crucial role in inducing interferon and cytokine production during infections with HSV, adenovirus, and poxvirus [38,39,40].

## 4. NOD-like Receptors

The NLR family is classified into four subfamilies (NLRA, NLRB, NLRC, and NLRP) based on the structure of their N-terminal effector domains. NLRs are expressed in the cytoplasm of immune cells such as macrophages and dendritic cells, as well as in non-immune cells, including epithelial cells, and play a crucial role in detecting molecules associated with intracellular infections [41,42]. NLRs recognize PAMPs such as those found in bacterial cell walls and danger-associated molecular patterns (DAMPs) released from injured cells [43]. The activation of these NLRs forms an inflammasome complex, which stimulates pro-interleukin-1β (IL-1β) and IL-18 production, triggering an immune response and contributing to host defense against pathogens [44].

NOD2 and NLRP9b in particular are thought to serve as viral nucleic acid sensors. NOD2 plays a key role in recognizing both bacterial-derived muramyl dipeptide motifs and virus-derived ssRNA, and subsequently activates downstream signaling pathways, including NF-κB and MAPK, leading to the induction of IFN-I [41,45,46,47]. NLRP9b recognizes viral dsRNA through the RNA helicase DHX9 and forms an inflammasome that inhibits rotavirus replication in intestinal cells [48].

## 5. RLRs, cGAS, ALRs, and DDX Molecules in the Antiviral Response

RLRs are the main sensors of infection by RNA viruses such as paramyxoviruses, flaviviruses, orthomyxoviruses, and coronaviruses, among others. These cytosolic sensors, which include RIG-I and MDA5, distinguish between self- and non-self-mRNAs by recognizing the PAMPs dsRNA with a 5′ di- or triphosphate or long dsRNA, respectively [49,50]. They then oligomerize and interact with a downstream scaffold protein associated with the mitochondrial membrane, Mitochondrial antiviral-signaling protein (MAVS), which initiates signaling via TRAF and the IKK family of proteins, resulting in the activation of IRF3, IRF7, and NF-kB. Another member of the RLR family, LGP2, also binds viral RNA but not MAVS and modulates RIG-I and MDA5 activity.

Retroviruses generate PAMPs at distinct stages of the replication cycle, which are recognized by different PRRs to orchestrate an effective antiviral response [51,52,53]. During retrovirus replication, host cells are exposed to various forms of nucleic acid, such as ssRNA, ssDNA, RNA–DNA hybrids, dsRNA, and dsDNA. These nucleic acids can be detected in the cytoplasm, and in some cases, within the nucleus when immature viral particles are internalized and degraded by host cells [54]. Consequently, multiple PRRs including TLRs, DDXs, ALRs, and cGAS are activated to initiate the host immune response. Furthermore, retroviral virions and proteins are recognized by PRRs [55]. It has also been suggested that lenti- and retroviruses incorporate the lipopolysaccharide (LPS)-binding protein CD14 into their membranes, which enables them to activate TLR4, the LPS PRR found on the cell surface [56,57].

cGAS is responsible for the innate immune response to cytosolic and nuclear DNA released from DNA viruses and retroviruses upon infection or produced in actively infected cells [58,59,60,61]. In response to binding DNA, cGAS synthesizes cyclic guanosine monophosphate–adenosine monophosphate (cGAMP) [58]. cGAMP is also produced when mitochondrial DNA released upon damage induced by RNA viruses such as influenza and flaviviruses binds to cGAS [62,63]. cGAMP binds to Stimulator of interferon genes (STING) on the endoplasmic reticulum (ER), triggering its activation and subsequent dimerization. Thereafter, STING associates with TANK-binding kinase 1 (TBK1) and translocates to the Golgi apparatus. TBK1 phosphorylates interferon regulatory factor (IRF) 3 and NF-κB, which then move to the nucleus, leading to the production of IFN-I and pro-inflammatory cytokines [64].

ALRs are a family of proteins with evidence of pathogen-driven strong positive selection that have important roles in innate immune signaling pathways [12,18,20,65,66]. ALRs consist of an N-terminal pyrin domain (PYD), which facilitates homotypic and heterotypic interactions with PYD-containing and other proteins, and a C-terminal hematopoietic expression, interferon-inducible nature, and nuclear localization (HIN) domain, which binds DNA [12,18]. *ALR* genes, which are encoded at a single locus in mammals, are under strong selection, and the copy number of these genes varies between species [18,65]. For example, in contrast to the 5 *ALR* genes found in humans, there are 13–16 mouse *Alrs*, depending on the inbred mouse strain, encoded at a single locus on mouse chromosome 1 [18,65,66,67].

AIM2 exhibits distinct characteristics from other ALRs and is classified into a separate clade in phylogenetic analyses; upon binding DNA, it activates the inflammasome pathway [12,18,65]. AIM2 is conserved across species in contrast to the other ALRs, which exhibit greater similarity to one another within the same species than across different species. The unique PYD and HIN domains of AIM2 compared to the domains found in other ALRs supports its different function [18,65]. Indeed, the HIN domains of the other ALRs, including IFI16 in humans and IFI203 and IFI204 in mice, bind DNA and induce IFN-I signaling in response to pathogens rather than inflammation [68,69,70,71,72].

IFI16 binds stem-rich ssDNA and viral dsDNA via its HIN-B domain to induce IFN-I responses but fails to induce IFN responses to RNA–DNA hybrids [69,73]. In addition to nucleic acid structures, it has been shown that nucleic acid length can influence the magnitude of the IFN-I response mediated by IFI16 [69]. This, combined with the fact that depletion of IFI16 leads to increased lentiviral replication, suggests that IFI16 has higher affinities for certain nucleic acid forms generated during viral infection [73]. Similarly, both cGAS and IFI16 are required for IRF-3 signaling in HIV-infected cells; cGAS especially plays a crucial role in the detection of HIV reverse transcription products and the dimerization of IRF-3 [74]. The murine ALR, IFI203, requires its HIN domain to bind early murine leukemia virus (MLV) reverse transcripts and strong stop ssDNA, and participates in the IFN-I response to MLV infection [68]. These studies demonstrate that different sequences, lengths, and secondary structures of pathogen nucleic acids might play a role in ALR pathogen recognition and the induction of innate immune responses.

In addition, IFI16 plays critical roles in the nucleus by orchestrating both innate immune sensing and epigenetic regulation of HSV DNA. IFI16, in cooperation with cGAS, detects HSV DNA as foreign, triggering innate immune signaling pathways that activate antiviral defense mechanisms. Additionally, IFI16 promotes the heterochromatinization of HSV DNA, leading to the epigenetic silencing of viral genes, thereby limiting viral replication. This function underscores IFI16’s role in integrating epigenetic regulation with innate immune responses, highlighting its importance as a nuclear defense mechanism that restricts HSV replication and enhances host immunity [75].

Paramyxoviruses may also activate the innate immune system via interaction with ALRs. Following infection with Nipah virus and measles virus, both cGAS and IFI16 activate STING, leading to the production of IFNβ [76]. Additionally, syncytium formation by paramyxoviruses results in the leakage of mitochondrial DNA into the cytoplasm, which is subsequently detected by cGAS and IFI16 [76].

A number of DEAD-box helicases in addition to the RLR family have been implicated in virus sensing. DDX3 recognizes short, abortive RNA transcripts produced in HIV-infected cells [77]. DDX41 has been implicated in the recognition of retroviral nucleic acids and may directly interact with STING following nucleic acid detection [65,66,68,69,78]. DDX41 is also a critical sensor of viruses in fish [79]. Like cGAS and several ALRs, nucleic acid sensing by DDX41 plays a crucial role in the activation of STING-dependent IFN induction, as well as the formation of inflammasomes and the initiation of inflammatory cell death [11,12]. Our group demonstrated that DDX41 detects the RNA–DNA hybrids generated during the first steps of MLV reverse transcription. Dendritic cells, but not myeloid-derived cells, were found to be primarily responsible for the in vivo control of viral infection by effectively initiating DDX41-mediated antiviral immune responses. DDX41 also recognizes RNA–DNA hybrids released from damaged mitochondria during influenza virus infection [62]. These findings suggest that DDX41 and cGAS recognize RNA hybrids and dsDNA produced at the early and later stages of reverse transcription, respectively [78].

## 6. The STING Pathway

The STING pathway is downstream of several nucleic acid sensors, including cGAS, ALRs, and DDXs. Recent studies have identified the degradation pathway for STING after it is activated, wherein ubiquitinated STING is targeted by hepatocyte growth factor-regulated tyrosine kinase substrate (HRS), a crucial component of the ESCRT-0 complex, resulting in its lysosomal degradation [80,81]. STING signaling is also terminated through the ESCRT-dependent microautophagy of vesicles originating from recycling endosomes [82].

There have been many reports regarding the recognition and regulation of DNA virus infection by the STING pathway [83]. STING pathway activation suppresses hepatitis B virus replication in human liver cell lines and in vivo mouse models [84], cytomegalovirus (CMV) replication in primary human endothelial cells [85], and HSV-1 replication in murine microglial cells [86]. DNA viruses have also evolved various strategies to antagonize the STING pathway and many evasion mechanisms and immunomodulators targeting the pathway have been identified [83]. For instance, the HSV-1 protein ICP27 interacts with the STING–TBK1 complex to inhibit type I IFN induction [87]. The UL82 protein of human CMV impairs the translocation of STING from the ER to perinuclear microsomes and inhibits the recruitment of TBK1 and IRF3 to STING [88]. The EBV large tegument protein BPLF1 is a virulence factor that mitigates cGAS-induced IFN-β production by antagonizing IFN-β gene transcription through its deubiquitinase activity [89]; and the pseudorabies virus tegument protein UL13 inhibits antiviral responses by recruiting the E3 ligase RING-finger protein 5 to induce K27/K29-linked ubiquitination and degradation of STING, thereby suppressing STING-dependent signaling [90].

We recently discovered that in mice, one member of the ALR family, IFI207 (PYHINA), increases STING signaling by stabilizing it [20]. The *Alr* locus in mice has experienced rapid evolution over the past few million years, leading to the emergence of two novel members, *MndaL* and *Ifi207*. Notably, *Ifi207* in particular exhibits substantial genetic variation even among closely related inbred mouse strains. We found that IFI207 is unique among ALRs due to a large repeat region separating the N-terminal PYD and C-terminal HIN domains that plays a crucial role in stabilizing STING. While IFI207/STING interaction did not affect the cytokine response to bacterial infections (*S. aureus*, *P. aeruginosa*, and *K. pneumoniae*), it contributed to the control of in vivo MLV infection. IFI207 enhanced the STING signaling pathway, leading to an increased antiviral response by stabilizing STING protein [20]. Whether humans or other species also encode ALRs or other factors that stabilize STING remains to be determined, although IFI16 has been reported to have the opposite effect of destabilizing STING [91].

## 7. The Role of Intracellular Nucleic Acid Sensors in Vaccine Development

The common mechanism underlying all effective vaccines is that the activation of innate immune responses serves as a crucial initiating event that alters the outcome of the adaptive immune response [92,93,94]. Vaccines are thought to utilize two primary types of immune triggers. The first involves PAMPs derived from the target pathogen, while the second pertains to vaccine components, such as specific adjuvants, that induce the release of endogenous DAMPs. Both PAMPs and DAMPs stimulate the innate immune system by activating PRRs [94]. Signals derived from PRRs are integrated at the level of antigen-presenting cells (APCs), thereby effectively modulating the adaptive immune response to the vaccine [94,95].

Recently, adjuvants have been developed that include additional substances that stimulate PRRs, such as the TLR3 agonist polyinosinic:polycytidylic acid (poly I:C) [7]. Moreover, defective interfering particles (DIPs), which are non-infectious virions that typically encapsidate subgenomic viral RNA molecules and are naturally found in measles virus, poliovirus, and influenza vaccine preparations, activate both TLR3 and TLR7 and are also being tested as adjuvants [7]. CpG oligodeoxynucleotide (CpG ODN 1018), a 22-mer sequence with a modified phosphorothioate backbone, is the only TLR9 agonist utilized in an approved vaccine for use in humans, specifically in the licensed hepatitis B vaccine [96,97]. Inactivated whole-virus influenza vaccines do not contain any added adjuvants, but the viral genomic RNA present in the vaccine formulation is believed to exhibit strong adjuvant activity, whereas this effect is weaker in inactivated or split vaccines [98]. The combination of the synthetic dsRNA, poly I:C, and monophosphoryl lipid A (a TLR4 agonist) is under development as an adjuvant for influenza vaccines [99]. Furthermore, cyclic dinucleotides (CDNs), including those activating cGAS, are expected to be developed as vaccine adjuvants due to their ability to induce safe, potent, and long-lasting humoral and cellular memory responses in both systemic and mucosal compartments [100].

DNA vaccines, which include the adjuvant effect of DNA plasmids to enhance immune responses, generate immunogens in vivo which are presented to the immune system, while also activating PRRs that respond to DNA molecules [101,102,103]. Specifically, upon DNA vaccination, Aim2 knockout mice exhibit significantly reduced levels of IFN-α/β, along with diminished humoral and cellular antigen-specific adaptive responses, indicating that the inflammatory responses induced by Aim2 play a crucial role in DNA vaccine efficacy [103].

Some studies have reported that the STING pathway plays a crucial role as an intracellular sensor for DNA vaccines. Notably, cytosolic double-stranded DNA strongly induces IFN-I in both immune and non-immune cells through the STING pathway, likely via interaction with cGAS. The pathway operates independently of traditional TLRs and CpG motifs, suggesting that STING is a key sensor for the innate immune response to DNA vaccines [103,104,105]. However, although it has been proposed that STING agonists might make good vaccine adjuvants, none are in clinical trials as of yet [106].

The advent of mRNA vaccines, particularly highlighted during the COVID-19 pandemic, has revolutionized vaccine technology by introducing viral genetic material into host cells to activate the innate immune system. The two key components of the currently approved mRNA vaccines for COVID-19 are nucleoside-modified mRNAs that encode the antigenic protein and lipid nanoparticles (iLNPs) containing ionizable lipids that facilitate the efficient delivery of intact mRNA to the cytoplasm of cells [3,4].

## 8. mRNA Vaccines and the Recognition Mechanisms of Non-Self RNA

mRNA vaccines can also stimulate innate immunity through TLRs (TLR3, 7, and 8) and the RLRs (RIG-I and MDA5) [107,108,109]. RIG-I recognizes ssRNA and dsRNA bearing a 5-triphosphates, stimulating IFN-I production [109,110]. In contrast, MDA5 detects long dsRNA generated during RNA virus replication, as well as synthetic RNAs, including poly I:C; the recognition of dsRNA by MDA5 activates IRF3 and NFκB, also leading to increased production of IFN-I [109,110] (Figure 2).

However, excessive activation of immune responses may compromise the safety of mRNA vaccines, highlighting the importance of understanding the distinctions between self and non-self mRNA structures. Notably, insights into this distinction have paved the way for RNA modification techniques [111]. Specifically, the recognition of non-self-nucleic acids is based on their structure, availability, and localization. Self-nucleic acids are typically degraded by nucleases before they can be sensed by nucleic acid receptors, preventing their recognition. In contrast, nucleic acid receptors recognize structural features characteristic of non-self RNAs, such as long dsRNA and 5′-triphosphate or 5′-diphosphate dsRNA, which trigger antiviral immune responses through these structural motifs [112]. In mRNA vaccines, modifications to structural elements such as the 5′ cap, 5′-and 3′-untranslated regions, coding region, and poly(A) tail prevent its recognition and degradation as non-self RNA by the immune system [113].

Furthermore, the dsRNA produced during in vitro is recognized in cells by PRRs such as TLR3 and RLRs, leading to the induction of IFN-I [106,107,108,109], as well as by protein kinase R leading to activation of the PKR–eIF2α pathway, which inhibits both cellular and viral translation, leading to apoptosis and thereby triggering the stress response [114]. Incorporation of pseudouridine and 2-thiouridine prevents the recognition of in vitro transcribed mRNA by TLR, RIG-I, and PKR, thereby improving vaccine effectiveness and minimizing the risk of inflammatory responses [115,116,117,118,119].

The mRNA cap structure is a major site of dynamic mRNA methylation and is classified into either Cap1 or Cap2 structures depending on whether the first transcribed nucleotide, or both the first and second transcribed nucleotides, are 2′-O-methylated [120]. The Cap1 structure on mRNA serves as an important marker for distinguishing self RNA from non-self RNAs such as viral RNAs which lack a cap, and by evading recognition by RLRs, viruses can effectively escape immune detection [121]. The dual methylation in Cap2 inhibits the binding of Cap1 to RIG-I, thereby suppressing the ability of endogenous RNA to activate the innate immune response [120].

In summary, these modifications when engineered into RNA vaccines enhance RNA stability, improve translation efficiency, and reduce the overall immunogenic response associated with RNA vaccines. While RNA modification techniques are important for enhancing RNA stability and modulating immunogenicity, the cytosolic sensors are also critical for ensuring the safety and efficacy of mRNA vaccines.

## 9. Conclusions

Cytosolic sensors regulate nucleic acid recognition and signaling pathways to appropriately control immune responses. Through these mechanisms, cells have evolved a multi-layered system to detect virus-derived components that serve as PAMPs, suppressing viral infection at an early stage via the innate immune system and facilitating the transition to adaptive immune responses. Thus, the mammalian innate immune system utilizes multiple PRRs to effectively control and eliminate viral infections, demonstrating sophisticated evolutionary adaptation.

Advancements in the understanding of cytosolic sensors and signaling mechanisms are essential for the development of effective and safe vaccines. Intracellular nucleic acid sensors are integral to the immunogenicity of live attenuated, killed, and nucleic acid-based vaccines. Their roles in mediating immune responses form the foundation for developing innovative strategies to enhance vaccine efficacy, facilitating the design of next-generation vaccines against infectious diseases and cancer. At the same time, the use of these molecules as adjuvants could lead to the excessive, uncontrolled activation of innate immune responses that may compromise vaccine safety. Future research is anticipated to further refine vaccine design based on these pathways, leading to more robust and sustained immune responses while minimizing untoward side effects.

## Figures and Tables

**Figure 1 vaccines-13-00193-f001:**
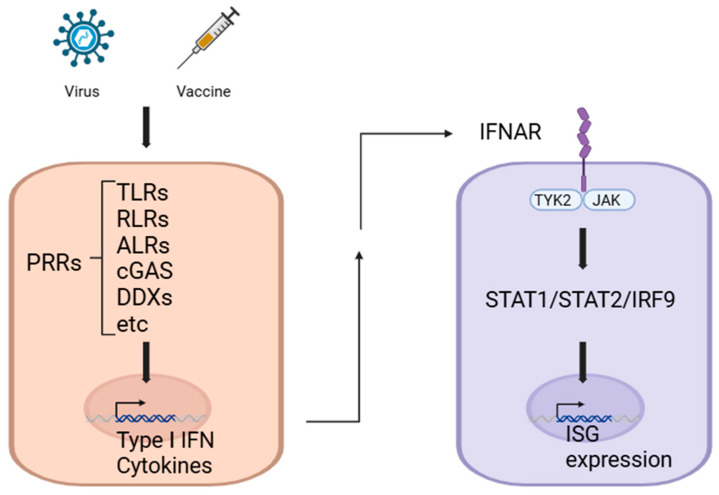
Key roles of pattern recognition receptors in viral defense and vaccine-induced immunity. The innate immune system utilizes PRRs to detect PAMPs and initiate rapid immune responses. As an example, PRR activation triggers IFN-I production, which through IFNAR signaling induces antiviral gene expression, enhancing viral defense and vaccine efficacy. Created in BioRender. ENYA, T. (2025) Accessed on 9 December 2024. https://BioRender.com/r45j084.

**Figure 2 vaccines-13-00193-f002:**
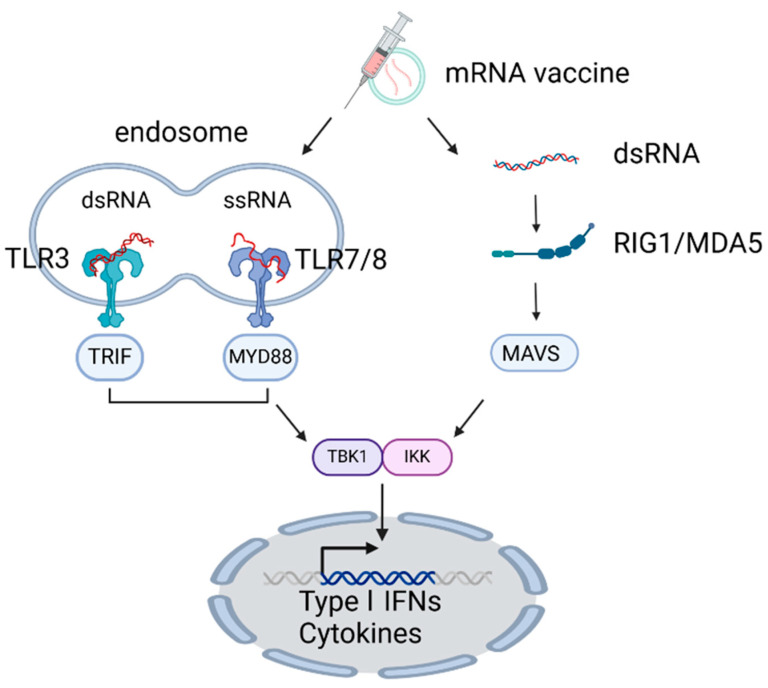
Mechanisms of innate immunity triggered by mRNA vaccine. mRNA vaccine activates innate immunity via TLRs (TLR3, 7, 8) and RLRs (RIG-I, MDA5), inducing IFN-I production through IRF3 and NFκB signaling. Created in BioRender. ENYA, T. (2025). Accessed on 9 December 2024. https://BioRender.com/p79c529.

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
