# Peer review of "Innate Sensing of Viral Nucleic Acids and Their Use in Antiviral Vaccine Development"

_vaccines, 2025, doi:10.3390/vaccines13020193_

Round 1
Reviewer 1 Report
Comments and Suggestions for Authors
This article is well-written and addresses an important topic in vaccine development- the recognition of viral nucleic acids by host mechanisms and the production of protective mechanisms such as tyoe 1 interferon and cytokines
The sections are logically divided and organized The new mRNA vaccines are also addressed.
I would suggest to expand discussion on epigenetic immunity and reference the following: Zhang J and Crumpacker C. HIV UTR, LTR, and Epigenetic Immunity,Viruses 2022; 14(5):1084
Author Response
This article is well-written and addresses an important topic in vaccine development- the recognition of viral nucleic acids by host mechanisms and the production of protective mechanisms such as type 1 interferon and cytokines. The sections are logically divided and organized the new mRNA vaccines are also addressed. I would suggest to expand discussion on epigenetic immunity and reference the following: Zhang J and Crumpacker C. HIV UTR, LTR, and Epigenetic Immunity,Viruses 2022; 14(5):1084
The Zhang and Crumpacker paper is very interesting but we were unable to incorporate it into our review, because the mechanisms by which epigenetic immunity works, as proposed by the authors, have little to do with nucleic acid sensing. We could not find a section in which to include this discussion.
Reviewer 2 Report
Comments and Suggestions for Authors
The authors provided a review on the importance of innate sensing of viral nucleic acids in anti-viral vaccine development. Their review explained how our immune system detects viruses and activates crucial responses to combat infections. By understanding these mechanisms, we can enhance vaccine efficacy and improve our defenses against viral diseases. This review discussed issues that are very significant for our understanding of the innate immune mechanisms and their contributions to vaccine development.
Here are some questions and suggestions:
1) LNPs are used for delivering mRNA into cells, and this LNP adjuvant is unrelated to immunity, so it seems unnecessary to discuss it in the text. (line 283-286)
2) Currently, there are some mRNA modification methods used to prevent mRNA vaccines from being recognized and cleared as non-self RNA. Could you briefly introduce this content in the text? This would deepen the readers' understanding of the relationship between innate immunity against non-self RNA and the development of mRNA vaccines.
Author Response
The authors provided a review on the importance of innate sensing of viral nucleic acids in anti-viral vaccine development. Their review explained how our immune system detects viruses and activates crucial responses to combat infections. By understanding these mechanisms, we can enhance vaccine efficacy and improve our defenses against viral diseases. This review discussed issues that are very significant for our understanding of the innate immune mechanisms and their contributions to vaccine development.
Here are some questions and suggestions:
1) LNPs are used for delivering mRNA into cells, and this LNP adjuvant is unrelated to immunity, so it seems unnecessary to discuss it in the text. (line 283-286)
As the reviewer correctly pointed out, iLNPs enable efficient delivery of intact mRNA to the cytoplasm of cells that can then translate the encoded protein. Therefore, we removed this from the Discussion.
2) Currently, there are some mRNA modification methods used to prevent mRNA vaccines from being recognized and cleared as non-self RNA. Could you briefly introduce this content in the text? This would deepen the readers' understanding of the relationship between innate immunity against non-self RNA and the development of mRNA vaccines.
We have added a description on mRNA modification methods that help avoid host recognition, including 5′ cap, 5′-and 3′-untranslated regions, coding region, and poly(A) tail (lines 339-346).
Reviewer 3 Report
Comments and Suggestions for Authors
This manuscript is well written. I have two suggestions that may improve the manuscript.
1. Section 3. The viral nucleic acid-sensing NOD-like receptors should be discussed, including NOD2 (PMID: 19701189), NLRP6 (PMID: 26494172, 34678144), NLRP9 (PMID: 28636595) etc.
2. Section 7. An important feature of mRNA vaccines is pseudouridine modification that minimizes PRR/PKR recognition while maximizes antigen expression. This should be discussed.
Author Response
This manuscript is well written. I have two suggestions that may improve the manuscript.
- Section 3. The viral nucleic acid-sensing NOD-like receptors should be discussed, including NOD2 (PMID: 19701189), NLRP6 (PMID: 26494172, 34678144), NLRP9 (PMID: 28636595) etc.
We have added Section 3 entitled “NOD-like receptors” (lines 114-129).
- Section 7. An important feature of mRNA vaccines is pseudouridine modification that minimizes PRR/PKR recognition while maximizes antigen expression. This should be discussed.
We have added a description of pseudouridine and 2-thiouridine modifications in mRNA vaccines (lines 323-329).
Reviewer 4 Report
Comments and Suggestions for Authors
Manuscript ID: vaccines-3388496, Review
Title: Importance of innate sensing of viral nucleic acids in anti-viral vaccine development
By Takuji Enya, Susan R Ross
The aim of this brief submission is to overview a contribution of innate sensing of viral nucleic acids in development of antiviral immune responses after vaccine application. This topic is timing. The growing body of evidence emphasizes the role of nucleic acids in induction of innate immunity, viral attenuation and protective immune responses.
After Introduction (1), in subsection 2, the authors overviewed common mechanisms of PRR (pattern recognition receptors) sensing and induction of innate immunity during retroviral infection. Retroviruses induce chronic/persistent infections and do not provide helpful lessons for better understanding antiviral innate and adaptive immunity induced by vaccination to prevent acute infections. Fig. 1 associated with this subsection is too simplistic and has limited visual value. In subsection 3, “The Role of Nucleic Acid Sensing in Pathogen Recognition and Immune Response”, the authors randomly picked-up some sensing and signaling pathways, e.g., ALR, cCAS, STING. These pathways are predominantly involved in recognition of DNA molecules. RIG-I-like receptors (RLRs) are critical for recognition of viral RNA of numerous human pathogens (influenza, YF, measles, CHIKV, Ebola, etc.). RIG-I and MDA5 receptors recognize different features of viral RNA molecules and LGP2 regulates RLR activation. Surprisingly, these receptors and RLR-dependent signaling pathways were not mentioned at all. The next subsection was totally dedicated to the STING pathway, mostly associated with recognition of foreign and/or damaged DNA molecules. In sub-section 5, “TLRs in Innate Immunity…”, the authors provided very limited update and contradictory interpretations. It is important to understand that TLR2-dependent cytokine production during LCMV infection noted by authors (line 231-233) is a feature of attenuated infection caused by mouse-adapted LCMV-ARM strain. Pathogenic “viscerotropic” LCMV-WE strain, and Lassa virus do not induce TLR2-dependent cytokine responses. In contrast, these arenaviruses inhibit TLR/Mal-dependent signaling and NF-kB-mediated proinflammatory cytokine responses. Accumulated evidence suggests that pathogenic arenaviruses affect RIG-I-dependent sensing and host innate immunity in a strain-specific manner. A growing body of evidence indicates that differences in terms of intracellular trafficking and increased interaction between LCMV-ARM and TLRs in the late endosomes can lead to more effective TLR- and RIG-I-based immune activation in contrast to pathogenic LCMV-WE (Johnson D et al., 2024).
Two short subsections, 6 (“The role of intracellular nucleic acid sensors in vaccine development”, and 7 (“mRNA Vaccines and the Recognition Mechanisms of Non-Self RNA“) are dealing with viral nucleic acids as vaccine adjuvants to improve vaccine efficacy. However, these sections are very sketchy and failed to provide solid analysis of recent publications in the field. The authors just listed some TLR agonists and mentioned their applications in some approved and experimental vaccines. There is no critical analysis of the positive and negative features of these agonists. Defective interfering particles, “inherited” potent nucleic acids adjuvants of existing live attenuated vaccines (influenza, measles, polio, IF17D), were only named. Most part of sub-section 7 was about the ability of RIG-I to discriminate between m7G-capped host RNA and foreign RNA with the 5’-triphospate modifications.
The conclusion section (8) is very limited (3 sentences) and does not provide justifiable directions for future studies.
In sum, this is a poorly written review that failed to provide an updated scientific assessment of viral nucleic acids as potent vaccine adjuvants for enhancement of innate immunity. The review is mostly based on papers published before 2020. Only 22% of references provided by the authors were published in 2020 and later. Viral nucleic acid sensing/signaling pathways is a very fast-growing area, and numerous very well written papers/reviews are currently available for experts and newcomers in the field (e.g., Zheng J et al., 2023; Lo R & Goncalses-Carneiro D, 2023; Cottrell KA et al., 2024; Gobbard AM et al., 2024; Solotchi M & Patel SS, 2024; Dorrity TJ et al., 2024; and etc.).
Author Response
The aim of this brief submission is to overview a contribution of innate sensing of viral nucleic acids in development of antiviral immune responses after vaccine application. This topic is timing. The growing body of evidence emphasizes the role of nucleic acids in induction of innate immunity, viral attenuation and protective immune responses.
After Introduction (1), in subsection 2, the authors overviewed common mechanisms of PRR (pattern recognition receptors) sensing and induction of innate immunity during retroviral infection. Retroviruses induce chronic/persistent infections and do not provide helpful lessons for better understanding antiviral innate and adaptive immunity induced by vaccination to prevent acute infections. Fig. 1 associated with this subsection is too simplistic and has limited visual value.
Fig. 1 was meant to illustrate the concept in both intracellular and bystander cell responses. Since the other 3 reviewers did not object to this figure, we prefer to retain it.
In subsection 3, “The Role of Nucleic Acid Sensing in Pathogen Recognition and Immune Response”, the authors randomly picked-up some sensing and signaling pathways, e.g., ALR, cCAS, STING. These pathways are predominantly involved in recognition of DNA molecules. RIG-I-like receptors (RLRs) are critical for recognition of viral RNA of numerous human pathogens (influenza, YF, measles, CHIKV, Ebola, etc.). RIG-I and MDA5 receptors recognize different features of viral RNA molecules and LGP2 regulates RLR activation. Surprisingly, these receptors and RLR-dependent signaling pathways were not mentioned at all.
These pathways were mentioned in the original manuscript (line 58 and beyond). With the reorganization, these are more prominently featured (lines 132-140).
The next subsection was totally dedicated to the STING pathway, mostly associated with recognition of foreign and/or damaged DNA molecules. In sub-section 5, “TLRs in Innate Immunity…”, the authors provided very limited update and contradictory interpretations. It is important to understand that TLR2-dependent cytokine production during LCMV infection noted by authors (line 231-233) is a feature of attenuated infection caused by mouse-adapted LCMV-ARM strain. Pathogenic “viscerotropic” LCMV-WE strain, and Lassa virus do not induce TLR2-dependent cytokine responses. In contrast, these arenaviruses inhibit TLR/Mal-dependent signaling and NF-kB-mediated proinflammatory cytokine responses. Accumulated evidence suggests that pathogenic arenaviruses affect RIG-I-dependent sensing and host innate immunity in a strain-specific manner. A growing body of evidence indicates that differences in terms of intracellular trafficking and increased interaction between LCMV-ARM and TLRs in the late endosomes can lead to more effective TLR- and RIG-I-based immune activation in contrast to pathogenic LCMV-WE (Johnson D et al., 2024).
There are many, many examples of viruses using the various pathways, and we are unable to provide details for all of these. As for the issue with LCMV, we now rephrase the sentence regarding TLR2 to read: “TLR2 also interacts with viral proteins as a TLR2/6 heterodimer, mediating cytokine production during infection with some strains of lymphocytic choriomeningitis virus (LCMV), New World arenaviruses, measles virus, respiratory syncytial virus, and herpes simplex virus (HSV)-1 [24–28].” (lines 100-103)
Two short subsections, 6 (“The role of intracellular nucleic acid sensors in vaccine development”, and 7 (“mRNA Vaccines and the Recognition Mechanisms of Non-Self RNA“) are dealing with viral nucleic acids as vaccine adjuvants to improve vaccine efficacy. However, these sections are very sketchy and failed to provide solid analysis of recent publications in the field. The authors just listed some TLR agonists and mentioned their applications in some approved and experimental vaccines. There is no critical analysis of the positive and negative features of these agonists. Defective interfering particles, “inherited” potent nucleic acids adjuvants of existing live attenuated vaccines (influenza, measles, polio, IF17D), were only named. Most part of sub-section 7 was about the ability of RIG-I to discriminate between m7G-capped host RNA and foreign RNA with the 5’-triphospate modifications.
We have expanded both these sections in response to the comments by several reviewers. We more fully explain what is meant by DIY (lines 271-273). We include a statement in the Conclusions section that discusses the need to ensure that nucleic acid vaccine adjuvants do not lead to hyper-stimulation.
The conclusion section (8) is very limited (3 sentences) and does not provide justifiable directions for future studies.
We have expanded the Conclusion section.
In sum, this is a poorly written review that failed to provide an updated scientific assessment of viral nucleic acids as potent vaccine adjuvants for enhancement of innate immunity. The review is mostly based on papers published before 2020. Only 22% of references provided by the authors were published in 2020 and later. Viral nucleic acid sensing/signaling pathways is a very fast-growing area, and numerous very well written papers/reviews are currently available for experts and newcomers in the field (e.g., Zheng J et al., 2023; Lo R & Goncalses-Carneiro D, 2023; Cottrell KA et al., 2024; Gobbard AM et al., 2024; Solotchi M & Patel SS, 2024; Dorrity TJ et al., 2024; and etc.).
We are sorry that this reviewer felt that the review is poorly written, although the other reviewers were of an opposite opinion. It is true that many of references we cited were prior to 2020; however, the fundamental pathways underlying nucleic acid sensing have been largely discovered prior to this date and much of the newer literature referred to by this reviewer pertains to the nuances of sensing of specific viruses. We have added 24 additional references, but since we are now at 120 references, feel that this is sufficient.
Round 2
Reviewer 4 Report
Comments and Suggestions for Authors
In the revised submission, authors modified subsections 3-5 describing TLRs, NOD-like and RLRs receptors and appropriate signaling pathways. Two sub-sections describing contribution of nucleic acids in vaccine-induced innate and adaptive immune responses were practically unchanged. As in the initial submission, the title of the paper is misleading. The major body of paper (lines 50-303) is about general mechanisms of innate immune nucleic acids sensing. There are numerous well-written reviews summarizing the latest development in this field. In the revised submission, only a little fraction of the paper (lines 305-348) dealing with contribution of vaccine nucleic acids in antiviral immunity. Major concerns raised by this reviewer were not properly addressed.
Author Response
In the revised submission, authors modified subsections 3-5 describing TLRs, NOD-like and RLRs receptors and appropriate signaling pathways. Two sub-sections describing contribution of nucleic acids in vaccine-induced innate and adaptive immune responses were practically unchanged. As in the initial submission, the title of the paper is misleading. The major body of paper (lines 50-303) is about general mechanisms of innate immune nucleic acids sensing. There are numerous well-written reviews summarizing the latest development in this field. In the revised submission, only a little fraction of the paper (lines 305-348) dealing with contribution of vaccine nucleic acids in antiviral immunity. Major concerns raised by this reviewer were not properly addressed.
In general, we disagree with this reviewer. We have read several recent reviews about nucleic acid adjuvants and feel that we are pretty complete in describing their use in anti-viral vaccines; the more recent reviews we saw also covered anti-cancer and other vaccines. Moreover, there are only a few vaccines currently in use (notably, the HBV vaccine containing CpG ODN 1018), which we had already described) that are in clinical use. Moreover, we are now up to 121 references, which suggests to us that we have covered a lot of the literature.
Nonetheless we have changed the title: “Innate sensing of viral nucleic acids and their use in anti-viral vaccine development” to hopefully satisfy their concerns.
The abstract, introduction and conclusion are very explicit about what we are trying to convey, so the reviewer is incorrect in stating that we do not talk about vaccine development except for the final section of the paper. Also, the reviewer is wrong – the final 2 sections as they mention in their critique (7. The role of intracellular nucleic acid sensors in vaccine development and 8. mRNA Vaccines and the Recognition Mechanisms of Non-Self RNA; lines 259 – 308) concerned nucleic acid sensing and vaccine development. We have added a few sentences to these sections and one additional reference.
It's pretty clear to us that we will not satisfy this reviewer, so we feel that the editor needs to make a decision at this point as to whether our manuscript is adequate.